# Evidence of Strong Guest–Host Interactions in Simvastatin Loaded in Mesoporous Silica MCM-41

**DOI:** 10.3390/pharmaceutics15051320

**Published:** 2023-04-22

**Authors:** Teresa Cordeiro, Inês Matos, Florence Danède, João C. Sotomayor, Isabel M. Fonseca, Marta C. Corvo, Madalena Dionísio, María Teresa Viciosa, Frédéric Affouard, Natália T. Correia

**Affiliations:** 1LAQV-REQUIMTE, Department of Chemistry, NOVA School of Science and Technology, Universidade Nova de Lisboa, 2829-516 Caparica, Portugal; teresa.m.cordeiro@gmail.com (T.C.); ines.matos@fct.unl.pt (I.M.); sotomayor@fct.unl.pt (J.C.S.); blo@fct.unl.pt (I.M.F.); madalena.dionisio@fct.unl.pt (M.D.); 2Univ. Lille, CNRS, INRAE, Centrale Lille, UMR 8207 - UMET - Unité Matériaux et Transformations, F-59000 Lille, France; florence.danede@univ-lille.fr (F.D.); frederic.affouard@univ-lille.fr (F.A.); 3i3N|Cenimat, Materials Science Department, NOVA School of Science and Technology, NOVA University, 2829-516 Caparica, Portugal; marta.corvo@fct.unl.pt; 4Centro de Química Estrutural, Institute of Molecular Sciences, Department of Chemical Engineering, Instituto Superior Técnico, Universidade de Lisboa, Av. Rovisco Pais, 1049-001 Lisbon, Portugal; teresaviciosa@ist.utl.pt

**Keywords:** simvastatin, amorphous state, molecular mobility, drug release, drug-carrier multiple interactions, drug delivery development

## Abstract

A rational design of drug delivery systems requires in-depth knowledge not only of the drug itself, in terms of physical state and molecular mobility, but also of how it is distributed among a carrier and its interactions with the host matrix. In this context, this work reports the behavior of simvastatin (SIM) loaded in mesoporous silica MCM-41 matrix (average pore diameter ~3.5 nm) accessed by a set of experimental techniques, evidencing that it exists in an amorphous state (X-ray diffraction, ssNMR, ATR-FTIR, and DSC). The most significant fraction of SIM molecules corresponds to a high thermal resistant population, as shown by thermogravimetry, and which interacts strongly with the MCM silanol groups, as revealed by ATR-FTIR analysis. These findings are supported by Molecular Dynamics (MD) simulations predicting that SIM molecules anchor to the inner pore wall through multiple hydrogen bonds. This anchored molecular fraction lacks a calorimetric and dielectric signature corresponding to a dynamically rigid population. Furthermore, differential scanning calorimetry showed a weak glass transition that is shifted to lower temperatures compared to bulk amorphous SIM. This accelerated molecular population is coherent with an in-pore fraction of molecules distinct from bulklike SIM, as highlighted by MD simulations. MCM-41 loading proved to be a suitable strategy for a long-term stabilization (at least three years) of simvastatin in the amorphous form, whose unanchored population releases at a much higher rate compared to the crystalline drug dissolution. Oppositely, the surface-attached molecules are kept entrapped inside pores even after long-term release assays.

## 1. Introduction

Simvastatin, [(1S,3R,7S,8S,8aR)-8-[2-[(2R,4R)-4-hydroxy-6-oxooxan-2-yl]ethyl]-3,7-dimethyl-1,2,3,7,8,8a-hexahydronaphthalen-1-yl]2,2-dimethylbutanoate, belonging to the antihyperlipidemic therapeutic class [1], is one of the most prescribed pharmaceutical drugs worldwide, including in the USA [2], and is used for the treatment of hypercholesterolemia [3,4] and dyslipidemia as an adjunct to diet.

Simvastatin (SIM) is a Class II drug according to the Biopharmaceutics Classification System. It is acceptably permeable [5] but is poorly water soluble [6] (7.25 × 10^−2^ mg L^−1^ at 25 °C in water [7], 24.4 mg L^−1^ in phosphate buffer at pH 6.8 and 37 °C after 48 h [8]), highly limiting its oral bioavailability (<5% [3,4]). Therefore, it would be beneficial if the solubility and/or dissolution rate were improved [9,10]. This is in line with the current goals of the pharmaceutical sciences and industry that seek to improve the effectiveness of already existing drugs through new formulations instead of investing in the design and synthesis of new molecules, which is costly and time-consuming [11].

In this context, alternative SIM administration routes were adopted [12,13,14], and several strategies have been addressed to enhance its solubility as drug particles reduction [15,16], solid lipid nanoparticles [17], solid surface dispersions [18], co-solvent evaporation with a hydrophilic polymer [19], emulsions and self-emulsifying systems [20,21], spray drying [22], and complexation [8,23].

Among the different approaches, amorphization has also been successfully used to improve SIM solubility [24]. Nevertheless, despite the relatively stabilization claimed for amorphous SIM due to strong hydrogen bonding [25] and high configurational entropy [26], it can convert to the stable crystalline counterpart depending on the way it was prepared [27], including particle size [28], or even to undergo chemical degradation as reported for mechano-activated amorphization [29]. Co-amorphization [30], solid dispersions [31], and the loading in glass nano-tube scaffolds [32] and mesoporous matrices [33,34] have been used to overpass the thermodynamic instability of the out-of-equilibrium amorphous form; however, recrystallization could occur depending on the pore diameter [34].

In the present study, amorphization of simvastatin (see structure in Figure 1) was achieved through incorporation in hexagonally ordered MCM-41 mesoporous silica with high specific surface area, pore volume and an average pore diameter ~3.5 nm, which was used earlier as a drug carrier [35]. These 100% silica matrices have the added benefit of low or no cytotoxicity [33,36,37,38,39]. The guest behavior was investigated by a set of experimental techniques: thermogravimetric analysis (TGA), Powder X-ray Diffraction (PXRD), solid-state Nuclear Magnetic Resonance (ssNMR), Attenuated Total Reflectance Fourier Transform InfraRed spectroscopy (ATR-FTIR), and Differential Scanning Calorimetry (DSC). Several studies in the literature report a significant solubility enhancement of amorphous SIM relative to its crystalline counterpart by a factor of two [28], five [24], ten [22], or even twelve thousand [23] times. Nevertheless, if amorphization is achieved by incorporation in a mesoporous matrix, its release could be slowed, or it can be retained by adsorption inside pores, acting as an indicator of the strength of guest–host interactions. In this context, SIM-MCM-41 interactions were thoroughly investigated by ATR-FTIR, and the way it is distributed in the matrix was inferred by DSC and molecular dynamics (MD) simulations. MD provided a picture of SIM distribution and anchoring inside pores, which, together with the molecular mobility of the loaded drug, are critical factors that could perform a role in the release mechanism [40]. The dynamical behavior of loaded SIM was probed by dielectric relaxation spectroscopy (DRS) that monitors reorientational motions of permanent dipoles under the influence of an external oscillating electrical field [41]. Studies providing the dielectric characterization of simvastatin-based delivery systems are scarce [34] and the same is true for the bulk amorphous drug [26,34].

The knowledge of the different relaxational modes active inside pores, SIM physical state, guest–host interactions, and distribution among the matrix will allow a better understanding of its release profile, contributing to a rational design of drug delivery systems.

## 2. Materials and Methods

### 2.1. Materials

Simvastatin (C_25_H_38_O_5_) was kindly gifted by Mepha, Lda. (CAS number 79902-63-9, >99% assay according to the supplier, molar mass of 418.6 g mol^−1^). It was used without further purification.

MCM-41 (100% Si) matrix, presenting an ordered mesoporous structure, was synthesized as described in reference [35].

### 2.2. Methods

#### 2.2.1. Matrix Textural Analysis

Nitrogen (N_2_) adsorption analysis through the N_2_ adsorption–desorption at −196 °C was used to determine the textural properties of the MCM-41 matrix before and after SIM loading. The specific surface area was determined with the Brunauer–Emmett–Teller (BET) method from the linear portion of the adsorption isotherm (relative pressures between 0.05 and 0.3 [42]). Total pore volume was estimated by calculating the amount of nitrogen adsorbed at the relative pressure P/P_0_ = 0.95 and the pore size distribution was determined by Brunauer–Joyner–Halenda (BJH) considering that pores are cylindrical and using the desorption branch data of the isotherm [43].

#### 2.2.2. Drug Loading

Prior to SIM loading, the mesoporous silica matrix (~150 mg of MCM-41) was submitted to heating at 150 °C, at vacuum pressure (10^−4^ bar) for 8 h, by immersion of the glass cell in a paraffin bath, to remove water and any impurity. After cooling to room temperature, a solution of 92 mg of simvastatin dissolved in 2 mL of chloroform was incorporated into the silica matrix under a vacuum. The solvent was evaporated under gentle stirring for 3 days at room temperature. The SIM-MCM-41 ratio used here was found after optimizing the loading conditions to avoid drug outside pores. Loaded SIM sample (designated as SIM:MCM) was prepared in a single batch allowing its characterization by all techniques.

#### 2.2.3. Thermogravimetric Analysis (TGA)

Thermogravimetric measurements (samples of ~1–4 mg) were completed from 22.5 °C to 550 °C, at a heating rate of 5 °C min^−1^, using a TGA Q500 apparatus (TA Instruments Inc., Guyancourt, France), under a highly-pure nitrogen atmosphere (sample purge flow rate of 60 mL min^−1^). The temperature and mass reading were calibrated using the Curie point of the nickel standard and using equibalance tare weights (provided by TA Instruments Inc), respectively.

The drug loading (*w*/*w*) percentage was calculated according to Equation (1).
(1)$$drug\;loading\;\left(w/w\right)\%=\frac{mass\;simvastatin}{mass\;simvastatin+mass\;silica\;matrix}\times 100\;$$

The filling degree (*v*/*v*) was estimated according to Equation (2), taking into account the loading percentage determined by thermogravimetry (Equation (1)) the total pore volume and density of crystalline simvastatin (1.172 g cm^−3^ [44]).
(2)$$filling\;\%=\frac{m_{SIM}\left(g\right)\rho \left({g\;\mathrm{cm}}^{-3}\right)}{pore\;volume\left({g\;\mathrm{cm}}^{-3}\right)m_{silica}\left(g\right)}\times 100$$

#### 2.2.4. Attenuated Total Reflectance-Fourier Transform Infrared (ATR-FTIR) Spectroscopy

ATR-FTIR spectra (400–4000 cm^−1^) were recorded using a Cary 630 FTIR spectrometer equipped with a diamond attenuated total reflectance, ATR (Agilent Technologies, Santa Clara, CA, USA), a dTGS detector thermoelectrically cooled, and KBr standard beam splitter. All spectra were recorded at room temperature via ATR method (resolution of 1 cm^−1^ and 16 scans). The spectrum corresponding to which is designated bulk amorphous SIM was collected after cooling a melted sample of crystalline SIM inside a DSC pan.

#### 2.2.5. Powder X-ray Diffraction (PXRD)

The X-ray diffraction analysis (5° < 2θ ≤ 60°; scan step = 0.0167 s/°) was performed at room temperature using a PANalytical X’Pert pro MPD diffractometer equipped with a Cu X-ray tube (λ_Cu Kα_ = 1.54056 Å) and the X’celerator detector. The powder samples were enclosed in a Lindemann glass capillary (diameter 0.7 mm), which was rotated during data collection.

#### 2.2.6. Solid-State Nuclear Magnetic Resonance (ssNMR)

Solid-state ^13^C MAS spectra were acquired at room temperature using a 7 T (300 MHz) spectrometer (AVANCE III Bruker) operating at 75 MHz (^13^C) and equipped with a BBO probehead. The samples were spun at the magic angle at a frequency of 5 kHz in 4 mm diameter rotors. ^13^C MAS NMR spectra were acquired with proton cross-polarization (CP/MAS) with a contact time of 1.2 ms, a recycle delay of 2.0 s, and the number of scans was 5000.

#### 2.2.7. Differential Scanning Calorimetry (DSC)

DSC measurements were performed using a DSC Q2000 operating in the Heat Flow T4P option and equipped with an RCS cooling system (TA Instruments Inc., Tzero DSC technology). The measurements were carried out under a very pure nitrogen atmosphere (flow rate of 50 mL min^−1^). Samples of ~5 mg were encapsulated in aluminum Tzero hermetic pans with a Tzero perforated lid to allow evaporation of water. The samples were first subjected to two cycles of cooling and heating (from −90 to 170 °C) at a rate of 10 °C min^−1^. Subsequently, a cooling and heating cycle was carried out at a rate of 30 °C min^−1^.

#### 2.2.8. Dielectric Relaxation Spectroscopy (DRS)

SIM:MCM composite and neat SIM were analyzed through dielectric relaxation spectroscopy. In short, for a material containing permanent molecular dipoles, the application of an external oscillating electrical field will cause the fluctuation of such dipoles with a preferential alignment, originating what is called orientational polarization. This is not an instantaneous response as it occurs in optical spectroscopies; instead, a phase lag exists between the stimulus—the variant electric field—and the build-up of orientational polarization. The stimulus-response relationship is described mathematically by the formalism of complex numbers [45], and therefore, a complex permittivity is defined as a function of an angular frequency, ω, ε*(ω) = ε′(ω) – *i*ε″(ω); the real component quantifies the energy stored by the material, and in the imaginary part, the energy is dissipated by the dipoles which can no longer follow the oscillating electric field [41,45].

A sample of each material, SIM and SIM:MCM, was sandwiched with two silica spacers (50 μm of thickness) in between two gold-plated electrodes of parallel plate capacitors. The sample cell (BDS 1200) was placed on a cryostat (BDS 1100), and the temperature (±0.5 °C) was controlled by Quatro Cryosystem. The real and imaginary components of the complex dielectric function, ε*(*f*) = ε′(*f*) − *i*ε″(*f*) (where *f* = ω/(2π) is the frequency of the applied oscillating electric field), were measured using an Alpha-N independence analyzer. Isothermal dielectric spectra, from 10^−1^ to 10^6^ Hz, were acquired over two runs, from −95 °C to 100 °C (first run) and from −95 °C to 165 °C (second run). All modules were supplied by Novocontrol Technologies GmbH.

#### 2.2.9. In Vitro Drug Dissolution and Drug Release Studies

A calibration curve was constructed with simvastatin solutions in concentrations ranging from 2 to 14 mg L^–1^, in phosphate buffer (5 mg L^–1^; pH = 6.8). A Thermo Scientific Evolution spectrometer and quartz cells were used for the acquisition of the absorption spectra over a wavelength range from 190 to 400 nm. The readings at 231, 238, and 247 nm were taken to build 3 calibration curves whose linearity (r^2^ > 0.99 for all the 3 chosen wavelengths) confirms Beer’s law applicability.

The dissolution assays of simvastatin were carried out by dissolving 2.85 mg of neat crystalline SIM in 200 mL of the phosphate-buffered solution. For the drug release trials, SIM:MCM was weighted to ensure that the sample contains 2.77 mg of SIM based on the percent loading (*w*/*w*) determined by TGA. Both neat SIM and composite were previously enclosed in a cellulose membrane SnakeSkin^®^ (cut-off 3500 Da) purchased from Thermo Scientific, allowing for the diffusion of SIM, but posing a barrier to silica particles. To closely mimic physiological conditions, the assays (three replicates) were performed at constant temperature in an incubator shaker (Comecta SA) regulated to 37 °C and 100 rpm.

#### 2.2.10. Molecular Dynamics Simulations

In the present work, Molecular dynamics (MD) simulations were performed using the DL_POLY package [46] to investigate SIM under confinement at the molecular scale. In order to model MCM-41 mesoporous matrices, we have used a fully pre-optimized configuration of vitreous silica provided in [47], in which the porosity was generated according to the procedure developed by Brodka and Zerda [48]. It allowed us to create a straight cylindrical channel of a mean diameter of about 3.5 nm carved in the pre-optimized amorphous silica (see Figure 1). The inner surface silanol density of about 7.5 OH nm^−2^ agrees with hydroxyl coverage in real highly hydrated silicate surfaces [49]. A set of 46 SIM molecules are confined in this straight cylindrical channel (see Figure 1), very similar to previous studies made on confined glycerol [50].

MD simulations were performed with a time step of 1 fs using periodic boundary conditions. We have used a semi-rigid model for the silica confinement matrix in which the position of silicon and oxygen atoms were fixed while the silanol hydrogens were allowed to rotate about the Si-O bond axis. In order to compute intra- and inter-molecular interactions, we have employed the all-atom OPLS force field [51] for SIM and the force field developed in [47,48] for silica. Van der Waals and electrostatic interactions between SIM and silica are computed using the Lorentz–Berthelot mixing rules, similar to previous studies [50]. The damped shifted method developed by Wolf was used to calculate coulombic interactions with the same cut-off radius of 10 Å used for both van der Waals.

All simulations of confined SIM were performed in the NVT ensemble (constant number of particles N, volume V, and temperature T). The Nosé-Hoover thermostat relaxation time has been chosen as 0.2 ps. First, one MD simulation of confined SIM has been performed at the temperature T = 600 K. At this high temperature, the system can be considered fully equilibrated over the duration of the MD simulation (150 ns). The diffusive regime is reached, and the time-dependent dipolar correlation function can be extracted. Second, in order to explore a lower and more realistic temperature, the confined SIM system has been rapidly cooled from 600 K to 300 K. This cooling procedure has been performed from ten different configurations obtained at T = 600 K over the MD simulation run. Then, ten independent MD simulations have been thus performed at T = 300 K based on the different states obtained after the repeated cooling processes. At T = 300 K, the system cannot be fully equilibrated, the diffusive regime is not reached, and the time-dependent dipolar correlation function cannot be extracted. However, this procedure involving multiple quenches allows us to obtain some insights into the possible structural organization of the liquid. Positions and dipole orientations of the SIM molecules can be extracted and averaged over the ten MD simulation runs in order to improve statistics.

The individual molecular dipole moments of SIM molecules have been calculated from the expression: $\vec{\mu }\left(t\right)=\sum_{\alpha }q_{\alpha }{\vec{r}}_{\alpha }\left(t\right)$ where $q_{\alpha }$ and ${\vec{r}}_{\alpha }\left(t\right)$ are the fixed charge localized on atom α and its position at time t, respectively. We have used a non-polarizable force-field, so fixed charges $q_{\alpha }$ are considered, which are directly taken from the OPLS force field [51]. The dipole moments of SIM molecules have been computed at the different investigated temperatures to obtain all dipolar properties of the system.

## 3. Results and Discussion

### 3.1. Textural Analysis

Figure 2a shows the obtained N_2_ adsorption (open symbols)-desorption (filled symbols) isotherms for the unloaded (grey curve) and loaded (blue curve) matrices. Unloaded MCM-41 presents a reversible type IV isotherm, as typically found for well-defined parallel mesoporous structures [52], exhibiting an inflection above (P/P_0_ > 0.9) due to the filling of interparticle porosities or voids. The obtained textural parameters with the Brunauer–Emmett–Teller (BET) method (specific surface area, total pore volume, and average pore diameter determined from BJH desorption branch (Figure 2b), are summarized in Table 1. After loading, the respective isotherm is highly modified revealing negligible nitrogen uptake similar to what is characteristic of a non-porous material with the specific surface area dramatically decreased (Table 1).

The type of isotherm for the loaded matrix is similar to the one found for other drugs loaded in mesoporous silica matrices, in which the filled pores are occluded in respect to N_2_ adsorption [53,54,55]. Since for the here studied composite, the average pore dimensions of MCM-41 (diameter ~3.5 nm) are larger than the molecular size of a SIM molecule (1.14 × 0.87 nm^2^; see inset in Figure 2b), which allows the drug to enter the pores, and therefore, the observed N_2_ pore occlusion is not caused by size exclusion (sieving [56,57]).

### 3.2. Thermogravimetric Analysis (TGA)

Thermogravimetric analysis was used to evaluate the resistance to thermal degradation of SIM when loaded in MCM-41 and simultaneously to determine the amount of drug loaded in the silica matrix. As will be shown in next section, SIM exists in the amorphous state in the composite (designated hereafter as SIM:MCM); thus, the monitoring of decomposition, besides the neat crystalline SIM, was compared with a previously amorphized sample obtained by cooling from melt (first heating run of the neat crystal up to 170 °C). All samples were submitted to heating from 22.5 °C to 550 °C at a rate of 5 °C min^−1^. Figure 3a shows the respective weight curves, including the one obtained for the unloaded matrix. Figure 3b shows the derivative plots for neat SIM and loaded matrix.

As a first observation, neat SIM decomposes according to a single step with an onset near 200 °C, and no significant differences occur between crystalline and amorphous samples. On the other side, the TGA curve for SIM:MCM follows a multi-step profile, better seen in the derivative plot (Figure 3b), where the different weight loss stages appear as peaks. After an initial rather small weight loss up to 100 °C due to water removal (1.1%), the weight loss of SIM in the composite occurs according to two steps. The onset of the first step is located at 100 °C, being associated with a fraction of less thermal resistant population ~4.2% (w_SIM_/w_composite_), meaning 12.6% (w_SIM_/w_SIM in the composite_). It is followed by a major degradation step with onset at 200 °C showing a greater contribution of a molecular population decomposing at higher temperature relative to neat SIM.

The unloaded matrix (grey curve in Figure 3a) after dehydration, shows no further weight loss in respect to temperature, as usually found for this type of matrix [35]. From Equation (1), after discounting the mass loss due to dehydration, the SIM loading is determined as 33.0% (*w*/*w*) (in agreement with the starting preparation), which, after conversion to volume (Equation (2)), gives a filling of 50.7% (*v*/*v*). The BET isotherm obtained for the loaded matrix (Figure 2a), thus corresponding, on average, to partial filling with pores occlusion. Owing to the bulkiness of each SIM molecule and the narrowness of the diameter of the pores, with some contribution of pore sizes below 3 nm (Figure 2b), only a small number of molecules are needed to obstruct the pores opening and filling the smaller pores. Therefore, the complementary information from the TGA and BET analysis indicates a distributed filling of the pores of the SIM:MCM composite.

### 3.3. Powder X-ray Diffraction Analysis

Figure 4 shows the powder X-ray diffraction pattern at room temperature of SIM:MCM in comparison with neat crystalline simvastatin. The diffractogram of crystalline SIM (black line) is characteristic of form I, assigned to the orthorhombic crystallographic system, space group, *P*2_1_2_1_2_1_ [44,58]. It is known that crystalline SIM can exist in three different polymorphs, among which form I is the stable one at room temperature; the other two polymorphs (form II and form III) are only detected at temperatures below room temperature, as first reported by Hušák et al. [59] (synchrotron powder diffraction and ss-NMR studies), and recently by Simões et al. [44]. In the diffractogram of SIM:MCM (blue line in Figure 4), no sharp Bragg peaks are observable, only a broad diffraction halo is detected, which proves that simvastatin is fully amorphous in the SIM:MCM composite.

### 3.4. ^13^C CP/MAS ssNMR Analysis

The ^13^C ssNMR spectrum of SIM:MCM is shown in Figure 5. All the ^13^C resonances are originated by SIM loading as it is possible to notice the absence of ^13^C chemical shifts in an unloaded MCM sample. The broadness exhibited in the loaded SIM ^13^C CP/MAS NMR resonances is similar to the spectrum for bulk amorphous SIM provided by Nunes et al., where distinct frequency regions can be found [26]. The complete assignment of ^13^C resonances of simvastatin has been first reported by Brus and Jegorov [60] (for crystalline form I and solution) and by Hušák et al. [59] (for the three simvastatin polymorphs). The carbonyl groups present a very broad signal between 180–170 ppm, the unsaturated carbons between 150–130 ppm, the C-OR moieties exhibit a broader signal than the previously described for bulk amorphous SIM, with resonances between 90–60 ppm, and the remaining aliphatic carbons appear between 50–10 ppm. The broadness of the carbonyl resonances (C1 and C18; see molecular structure in Figure 1) supports the existence of a distribution of conformations and the lack of order at shorter distances. Moreover, the observation of broader and downshifted frequency groups is compatible not only with the existence of an amorphous state but also with the establishment of possible SIM-MCM interactions.

### 3.5. ATR-FTIR Analysis

To get further information about the drug’s physical state, ATR-FTIR spectroscopy was used, which also allows for obtaining insights into the SIM intermolecular interactions. The spectrum corresponding to bulk amorphous SIM (see Section 2) was collected in addition to the one of neat crystalline SIM (form I). In this phase, infinite chains along the crystallographic $\vec{b}$ axis are formed due to intermolecular single hydrogen bonds that establish between the hydrogen of the hydroxyl group of the lactone ring (see Figure 1) of one SIM molecule and the oxygen of carbonyl in the linear butyl ester [44,58] of a second SIM molecule (O3−H···O5). Therefore, the spectroscopic analysis will be focused on the two infrared regions where vibrational modes of O−H (between around 3700 and 3100 cm^−1^) and C=O (between 1800 and 1600 cm^−1^) absorb. Free O−H and C=O have characteristic frequencies at approximately 3520 and 1760 cm^−1^, appearing as sharp lines [61]. All spectra are compared in Figure 6.

The spectrum of neat crystalline SIM in the O−H region is shown in Figure 6a, displaying a sharp band centered at 3547 cm^−1^ due to OH hydrogen-bonded (O3−H···O5) [30]. Upon amorphization, the global spectrum shifts towards a lower wavenumber region, becoming broader (Figure 6b), consistent with what is reported in the literature for the bulk amorphous SIM [27,28,30] due to its less organized structure, evidencing that no free OH groups exist, which would give rise to infrared signals at higher wavenumbers. Therefore, it can be concluded that in the amorphous form, the hydroxyl groups of the lactone ring of all SIM molecules are interacting via hydrogen bonding (HB). This HB interaction of an OH group of a SIM molecule can be established with another OH group (O3−H···O3−H) and/or with the carbonyl group of the ester tail (O3−H···O5) and/or of the lactone ring (O3−H···O2) of a second SIM molecule; see SIM chemical structure and atoms numbering in Figure 1.

The profile of the O−H band of SIM incorporated in MCM-41 (Figure 6c) clearly shows no sign of crystalline phase and differentiates from the one of bulk amorphous SIM, having an additional contribution in the low wavenumber region highlighted by the shadowed area. This additional contribution acts as a clear indication that hydrogen bond interactions between the OH groups of SIM molecules and the silanol (Si-OH) pore wall groups exist. Due to their location at even lower wavenumbers, it is possible to conclude that these guest–host interactions are stronger than those between SIM molecules. Note that the modification of the O−H band comes neither from the matrix itself nor from water vibrations. In fact, the unloaded MCM-41 spectrum is included in Figure 6c (grey line), for which the water content is similar to that of the composite (1.6%(*w*/*w*)), being clear that it has a negligible influence on the composite infrared spectrum.

The infrared spectrum of neat crystalline simvastatin in the carbonyl stretching region shows a structured band (Figure 6d) that can be deconvoluted in three contributions (1725, 1710, and 1695.4 cm^−1^). Details on deconvolution and bands assignment are given in the Appendix A (Figure A1). The band at 1695.4 cm^−1^ is assigned to the stretching of hydrogen-bonded carbonyl in the butyl ester tail, C=O5_ester,HB_ (1695 cm^−1^ [29]/1693.5 cm^−1^ [30]) which ensures the infinite one-dimensional chain along the crystallographic $\vec{b}$ axis: O3−H···O5 [44,58]. The remaining two bands, that were also identified in the literature (1720 and 1708 cm^−1^) [29], are assigned to stretching modes of the lactone carbonyl group, C=O2_lact_, which is involved in weak HB interactions, C9−H···O2 and C2−H···O2, ensuring the 2D and 3D molecular packing in crystalline SIM [44]. In bulk amorphous SIM, the 1750–1500 cm^−1^ spectral region (Figure 6e) is modified according to what is already reported in the literature [28,29,30]. The shift to higher wavenumbers of the global band suggests that, at least partially, carbonyl groups of the ester butyl tail become weakly or free HB in the amorphous state.

In the SIM:MCM composite (Figure 6f), the global band of carbonyl absorption becomes less structured, being centered at a slightly lower wavenumber than that of bulk amorphous SIM, with an additional low wavenumber contribution, highlighted by the shadowed area. The appearance of a low wavenumber contribution indicates that SIM molecules are strongly interacting with the silanol groups of the MCM-41 pore wall via the carbonyl groups (O5···H−O−Si and/or O2···H−O−Si) in addition to the hydroxyl-silanol interactions revealed by the analysis of the O−H vibration band (Figure 6c).

ATR-FTIR analysis of the SIM:MCM composite clearly indicates the existence of a fraction of simvastatin molecules strongly interacting with the inner walls of MCM-41 pores, consistent with the high thermal resistance population evidenced by TGA analysis.

### 3.6. Differential Scanning Calorimetry (DSC) Analysis

PXRD, ssNMR, and ATR-FTIR analyses at room temperature gave evidence that SIM loaded in MCM-41 is in an amorphous state. To better understand its physical state and thermal behavior, calorimetric measurements were performed in a wide temperature range for SIM:MCM composite in comparison with neat crystalline SIM and bulk amorphous SIM.

Figure 7a depicts the DSC curves collected in the first heating at 10 °C min^−1^ for neat crystalline SIM (black line) and SIM:MCM composite (blue line). The sharp melting endotherm centered at T_m_ = 141 °C (T_m,onset_ = 140 °C) with an associated enthalpy of ∆H_m_ = 74.14 J g^−1^ (31.0 kJ mol^−1^) is characteristic of Form I, in close agreement with which is reported in the literature [26,62]. The absence of a melting peak in the DSC curve collected in the first heating run for SIM:MCM (blue line in Figure 7a) confirms that no crystalline SIM exists in the composite, in agreement with the spectroscopic analyses (ATR-FTIR and ssNMR) and PXRD, in which only the amorphous halo is detected. The absence of melting indicates that no drug exists outside the pores, contrary to what was observed for a different batch prepared with a higher SIM/MCM weight ratio (see additional DSC results in Appendix A, Figure A2).

In the DSC curve (Figure 7a), a broad endotherm due to water removal is detected, making it difficult to observe, in this first run, a heat capacity step associated with the glass transition of amorphous SIM in the composite. From the weight loss registered at the end of the DSC measurement, a 2.0% (*w*/*w*) was determined for the water amount in the composite, in close agreement with TGA quantification, despite the long period of time between the different analyses. This low tendency of SIM:MCM composite for water uptake is compatible with the picture of SIM molecules adsorbed at the pores entrance, impairing the incoming of water molecules, which occurs for nitrogen in the BET assay shown in Figure 2a.

After water evaporation, a low intense discontinuity associated with the glass transition is detected in the following curve at 10 °C min^−1^ (not shown). To increase the steepness of the heat flow discontinuity at the glass transition temperature (T_g_), the DSC curve was acquired with a higher heating rate (30 °C min^−1^), better revealing the glass transition step (blue curve in Figure 7b). The respective onset is located at a temperature inferior to the one of bulk amorphous SIM, carried out at the same heating rate (blue vs. green curve in Figure 7b). The glass transition becomes also evident in a heat flow derivative representation against temperature, where it emerges as a downward peak; this is shown in Figure 7c for both composite and bulk SIM. The obtained profile of confined SIM points to the inexistence of a bulk amorphous fraction outside the pores, contrary to what was observed for a different batch prepared with a higher SIM/MCM ratio (Figure A2 in Appendix A). A T_g_ decrease of ~15 degrees relative to bulk amorphous SIM is observed, as seen by the difference between the peak’s minima in the derivative plot. Trying to quantify this highly mobile population, the heat capacity variation at T_g_, ΔC_p,SIM:MCM_ was compared with the respective heat capacity of bulk SIM, ΔC_p,SIM_ after correction for the SIM mass in the dehydrated composite and taking into account the loading degree (33% (*w*/*w*)). The obtained ratio (ΔC_p,SIM:MCM/_ΔC_p,SIM_ × 100) gives ~13.7% of the total SIM population in the SIM:MCM composite, corresponding to 4.5% (*w*/*w*) in relation to the total composite mass, in good agreement with the 4.2% (*w*/*w*) less thermal resistant SIM fraction as quantified by TGA.

The onset of the glass transition at lower temperatures relative to bulk SIM, and the absence of an enthalpy overshoot, which appears superimposed on the glass transition heat flux step for bulk amorphous SIM (positive peak in the bulk SIM derivative plot shown in Figure 7c), are features also found for ibuprofen incorporated with an analogous filling in a similar MCM-41 matrix (average pore diameter 3.5 nm) [35]. This was interpreted as a consequence of confinement effects [63,64] manifesting when the pore dimensions of the host matrix interfere with the drug cooperative length scale, being originated by an accelerated pore-core population. In the ibuprofen-loaded MCM-41, the glass transition is rather broadened compared with the bulk drug, extending to temperatures above the bulk one. This is due to a population inside pores that gradually distributes from the accelerated molecules in the pore core, with no interactions with the silica surface, to a high-T hindered population associated with pore-wall adsorbed ibuprofen molecules, however, keeping some limited mobility. Differently, in SIM:MCM composite, only the decreased glass transition contribution is clearly found with a lack of a calorimetric signature of the remaining loaded population whose existence has been clearly demonstrated by the thermogravimetric analysis. Therefore, the non-existence of a change in heat capacity at high temperatures indicates a highly constrained molecular population impeded in terms of its ability to undergo configurational rearrangements as a consequence of strong adsorption on the silica matrix. This was evidenced by ATR-FTIR, and it is coherent with an extended high thermally resistant fraction found by TGA.

To get further insight in the dynamical behavior of SIM confined in the composite, measurements by dielectric relaxation spectroscopy were carried out and compared with bulk amorphous SIM.

### 3.7. Dielectric Analysis

Dielectric relaxation spectra of SIM:MCM were collected isothermally in the temperature range from −95 °C to 165 °C after a prior run carried out up to 100 °C to assure water release (see details in Appendix A, Figure A3). Note that the MCM-41 matrix itself (dried) does not contribute to the dielectric spectrum of the composite [35]. The comparison of the dielectric response of loaded simvastatin (after dehydration) with bulk amorphous SIM is shown in Figure 8.

Figure 8a depicts the isochronal plots of the imaginary component of the complex permittivity (ε″(T)) at four different frequencies, between 10^2^ and 10^5^ Hz. Bulk amorphous SIM presents two relaxation processes associated with local mobility (named *γ* and *β* in increasing order of temperature) at the lowest temperatures, and, above ~25 °C, the signature of the high-intense process associated with the dynamic glass transition (*α*); at the highest temperatures, the spectra are influenced by direct current conductivity (*σ_dc_*), removed from main figure for clarity, but included in the inset. The detected processes agree with the literature reports [26,34]. As the main dipolar moieties, in a SIM molecule, are the hydroxyl and ester butyl tail (see Figure 1), the dielectric secondary relaxation processes are associated with fluctuations of these polar groups as assigned in a joint study by DRS and solid-state nuclear magnetic resonance spectroscopy [26]. The latter technique gives evidence that the polar hydroxyl group is the one presenting the higher resonance frequency. Since the OH group is the more mobile one, its reorientational motions should be associated with the γ-process, the one dielectrically detected at the highest frequencies (faster relaxation rate), which, in an isochronal plot, emerges at the lowest temperatures. While this process is noticed in the ε″(T)-trace of bulk amorphous SIM (inset of Figure 8a), it is not so clearly seen in the composite being probably highly depleted. As the ATR-FTIR spectra showed, both bulk and loaded SIM have no HB-free OH groups. However, in SIM:MCM composite, a fraction of OH groups are hydrogen bonded to the matrix silanol moieties (low wavenumber region highlighted in Figure 6c) instead of other SIM molecules as occurs in the bulk drug. Therefore, the weak dielectric response of the *γ*-relaxation (Figure 8c) in SIM:MCM reinforces the infrared analysis, which points to the existence of a fraction of molecules strongly anchored to the pore wall hindering the OH group mobility.

Concerning the *β*-process, it is clearly noticed in the isochronal plot of SIM:MCM composite (the low-temperature region in Figure 8a), keeping the same location and overall shape as the respective process detected for bulk SIM. The resemblance of the dielectric response is also shown by the isothermal representation at −50 °C and −20 °C (see Figure 8b). Figure 8c presents the spectral temperature evolution in SIM:MCM evidencing that the same temperature increment provokes equal frequency shift in the loss peak of the *β*-relaxation (this type of change in peak position with temperature is characteristic of a thermally activated process). The inset of Figure 8c depicts the predicted linear dependence against 1/T of the logarithm of the respective relaxation time (*τ*), estimated from the peak maxima in an isochronal representation (Figure 8a; the maximum temperature of the *β*-peak in the ε″(T)-trace, T_max_, is taken for each frequency, *f* and *τ* = 1/(2π*f*)). The fitting by the Arrhenius law, τ(T) = τ_∞_exp(E_a_/RT) (R (ideal gas constant) = 8.3145 J mol^−1^ K^−1^), gives an activation energy of 47 kJ mol^−1^ for SIM:MCM, close to the corresponding value in bulk SIM, 52 kJ mol^−1^, and in good agreement with the values reported in the literature [34]. This Arrhenius temperature dependence is characteristic of local, intramolecular secondary relaxations, which should be attributed to dipolar fluctuations in the ester tail, in line with solid-state NMR observations [26]. The dipolar moieties contributing to the detected *β*-process in SIM:MCM must correspond to a fraction of the butyl ester groups that are non-interacting with the matrix and, thus, have comparable mobility to the homologous intramolecular process in bulk amorphous SIM. This points to the sensitivity of DRS for probing short-length scale motions in a complex system in which only a small fraction of the sample responds dielectrically.

Figure 8c also evidences the nonappearance of a cooperative bulk-like *α*-process in the low-frequency tail of the *β*-relaxation in SIM:MCM, unlike what happens in the bulk SIM, as illustrated in the isotherm at 10 °C shown in Figure 8d. This is in line with what was found by DSC, where no significant heat flux discontinuity associated with a bulk glass transition was detected; instead, a small C_p_ jump at a lower temperature was found. The latter, in dielectric measurements, should correspond to a weak *α*_fast_-relaxation and, therefore, is expected to appear at lower temperatures/higher frequencies relative to the SIM bulk *α*-process, approaching and probably submerging under the bulk-like *β*-process. However, no significant change in the shape of the dielectric loss peak of the *β*-process is observed, as one would expect if such a coupling of relaxation processes of a different nature (intermolecular versus intramolecular, respectively, for the *α* and *β*-process) has occurred. Therefore, we can hypothesize that the molecular arrangement of SIM molecules inside the core of the pores leads to a strong decrease or even total cancellation of the global relaxing dipole moment weakening the corresponding *α*_fast_-relaxation dielectric response. This feature will be further elucidated by MD simulation studies (Section 3.9).

It is worth mentioning that recrystallization was not observed for loaded SIM, even after 3 years of storage at ambient conditions. This, in addition to the evidence that no recrystallization was observed when SIM:MCM was subjected to thermal treatments (DSC and DRS), can be taken as an indication that the critical nuclei diameter for crystallization is higher than 3.5 nm and, therefore, no driving force exists to trigger crystallization when SIM is confined to such narrow pore size.

Additionally, in the isochronal plot of SIM:MCM composite (Figure 8a), a high-temperature tail emerges that needs clarification. Taking other low molecular weight glass formers loaded in similar matrices as reference [63,65,66], they typically exhibit two high-temperature processes: (I) a surface one due to the relaxation of weakly adsorbed molecules at the inner pore wall, which are still mobile but with reduced mobility compared to non-adsorbed molecules; and (II) an interfacial polarization (Maxwell–Wagner–Sillars (MWS) process [67]), characteristic of disordered inhomogeneous media where dielectrically different material coexists. While process (I) gives a high temperature glass transition signature in DSC, the MWS process has no calorimetric response. Given that in SIM:MCM composite, no signal is clearly distinguished by the calorimetric analysis, other than the low-T glass transition, as already commented, it seems reasonable to attribute the dielectric peak, emerging at high-temperatures, to interfacial polarization building up at the guest–host interfaces. Furthermore, also some SIM evaporation could contribute, above 100 °C, leading to an apparent frequency-independent peak in the ε″-trace at ~130 °C. This possibility just gains relevance in DRS isothermal spectral acquisition, carried out here in steps of 2–5 °C, with an equivalent heating rate between 0.4 °C and 0.6 °C min^−1^ (much lower than the heating rate practiced in the DSC analysis), which may give time for evaporation of a small SIM fraction.

The non-detection of a dielectric and calorimetric high-T surface process, indicating that the associated population of SIM molecules are dynamically rigid (in the temperature range explored) provides further evidence that these molecules interact strongly with the host surface, corroborating the information from infrared and thermogravimetry analyses.

### 3.8. In Vitro Drug Release Studies

Simvastatin release from the MCM carrier was monitored by UV-Vis spectroscopy in a phosphate-buffered solution (pH = 6.8) at 37 °C and 100 rpm, to mimic intestinal fluid; crystalline simvastatin’s dissolution was followed under the same conditions. The obtained profiles are shown in Figure 9, after normalization by the total amount of SIM used in the respective assay: 2.9 mg/200 mL solution for neat crystalline SIM and 8.5 mg composite/200 mL solution (corresponding to 2.8 mg of amorphous SIM loaded in MCM as estimated from the TGA analysis), after averaging over three trials.

As displayed in Figure 9a, a completely different profile was obtained for the SIM release from MCM carrier (blue circles) as compared with the crystalline drug dissolution (grey circles). Indeed, an almost insignificant dissolution of the latter occurs in the first 6 h, as scaled up in Figure 9b, and only after 30 h does it starts to progressively dissolve, attaining completion after 56 days (1344 h). This behavior reflects the very low dissolution rate of such poorly water-soluble drug. It is important to note that trials were conducted at ~13 mg L^−1^ of SIM in either crystalline state (dissolution assays) and amorphous in SIM:MCM (release assays), a concentration lying below the solubility limit of crystalline SIM, 24.4 mg L^−1^ as determined under the same pH and temperature conditions [8]. 

By other side, in the same initial period, a significant amount of SIM is released from the MCM carrier, evidencing how drug amorphization leads to the enhancement of dissolution rate relative to crystalline SIM. A ~12% SIM_released_/SIM_total_ fraction is delivered in the first 3 h from the SIM:MCM composite, as denoted by the first stage in the release profile (see Figure 9b). This amount compares to the less thermal-resistant population quantified by TGA (conversion from 4.2% w/w_composite_ to 12.6% w/w_SIM_) and to the heat capacity step registered in the DSC curve (13.7% w/w_SIM_). Therefore, the first fraction released should correspond to the delivery of SIM molecules with increased mobility (low T_g_) as provided by DSC and diffusing from the center of the pores. Subsequently, some adsorbed molecules, most likely not strongly interacting with the pore walls, are released. Nevertheless, the composite SIM:MCM never attains full release, the maximum percentage being 39.6% SIM_released_/SIM_total,_ indicating that a significant fraction of population (~60%) remains trapped inside pores. This is due to the strong attraction of SIM molecules with the host surface, as highlighted in previous sections. Moreover, as will be seen in the next section (molecular dynamics simulation studies), the fact that SIM interacts with the inner pore walls via multiple hydrogen bonds involving the hydroxyl and carbonyl groups of the lactone ring and also the carbonyl of ester tail, leaves a hydrophobic molecular domain facing the center of the pores, an effect that is even more striking for the configuration in which SIM is simultaneously attached to the pore wall at five sites. This hydrophilicity reduction decreases surface wettability, impairing the entrance of the aqueous-releasing media and, consequently, the release of the pore-wall attached drug.

### 3.9. Molecular Dynamics Simulations

Molecular dynamics (MD) simulation studies (see details in the Methods Section) were performed to investigate the structural and dynamical properties of the SIM molecules inside MCM-41 pores. The simulated pore diameter (Figure 1) is close to the average pore size ~3.5 nm given by BJH desorption branch for the MCM-41 carrier used in the present work, which has a very narrow pore size distribution (see Figure 2b).

Figure 10 shows the radial density function of SIM molecules along the pore radius r, at T = 300 and 600 K. The density profiles were computed by considering the center of mass of SIM molecules. The effects of the confining matrix on the fluid structure are clearly emphasized by the overall profile.

At r > 10 Å, the presence of a broad and intense contribution demonstrates that the molecules organize by forming a clear surface layer due to the interaction of the SIM molecules with the surface. The formation of strong hydrogen bonds between the SIM molecules and the silanol groups on the silica surface is clearly suspected, as displayed in Figure 11. Such a surface-induced layering has already been observed from MD simulations in many systems [50,68]. Interestingly, this intense peak at radius larger than about 10 Å (Figure 10) exhibits two components, which could originate from the large size and non-globular shape of the SIM molecule (see Figure 1 and inset of Figure 2b) characterized by at least two characteristic dimensions 8.7 Å to 11.4 Å. In Figure 11b, the center of mass of the SIM molecules is closer to the pore surface than in Figure 11a, which could also explain the existence of two components in the broad peak of radial density function at r > 10 Å (Figure 10). It could be noted that the overall shape of the radial density profile (peaks position and maxima intensity) close to the surface r > 10 Å is weakly affected by temperature, suggesting that this surface layer exists at all temperatures. At distances r < 10 Å, the modulation of the radial density is less marked and vary around the expected average density of SIM in the channel (0.94 g cm^−3^). At T = 600 K, one may notice a bump at r = 6 Å which could indicate the presence of a second layer. At T = 300 K, the presence of multiple visible broad but distinct peaks may reveal a structuration of the liquid as the temperature is lowered. SIM molecules cannot be thus considered as really bulk-like in the central part of the pore oppositely to the trend observed for smaller molecules, such as water ([50,68,69]), confined in a similar type of mesoporous matrix. This could originate from the large size of the SIM molecule (see inset of Figure 2b) with characteristic dimensions ranging from 8.7 Å to 11.4 Å to be compared to the pore size of 35 Å. Interestingly, the presence of a clear minimum in the radial density function at a distance close to 10 Å allows us to clearly distinguish one inner zone of the pore (i.e., the central part of the pore or pore volume) at r < 10 Å and one outer zone of the pore close to the surface at r > 10 Å (pore surface). The possibility to clearly distinguish these two zones will be employed in the following.

Figure 12 provides some insights of the orientation of the SIM molecules inside the mesopore. It shows the distribution P(ϕ) of the angles ϕ of the SIM molecules, for which ϕ is the angle between the dipole moment of the SIM molecules and the vectors normal to the pore surface (see Figure 2). The function P(ϕ) has been computed for the SIM molecules in the pore volume and the pore surface zones and at the two investigated temperatures 300 K and 600 K. Interestingly, Figure 12 reveals that dipoles are not randomly distributed inside the mesopore but show a preferential orientation (ϕ≈0). Dipole moments are thus preferentially pointing toward the pore surface. This behavior is consistent with molecules participating in some H-bonds with SIM interfacial oxygens and hydrogen atoms belonging to the pore itself (see Figure 11). This trend is very marked for the dipoles close to the pore surface, i.e., for molecules in direct interaction with the surface, but it is even also observed for the dipoles in the pore volume, although the effect is less important. Actually, it confirms that SIM molecules are not bulk-like, even in the central part of the mesopore. The alignment of the SIM dipoles toward the surface also seems to amplify upon decreasing the temperature, as shown by the increase in the P(ϕ≈0) intensity from 600 K to 300 K. This strong preferred orientation of SIM dipoles in the pore (especially in the vicinity of the surface) which differs to the bulk phase where SIM dipoles do not exhibit any preferential orientation may have a strong consequence for all dielectric properties observed in confinement. One may thus expect a strong decrease in the dielectric strength because dipoles cancel each other’s due to their radial orientation.

These results of MD simulations clearly explain the non-detection by dielectric relaxation spectroscopy of an α_fast_ process associated with the SIM molecular population with accelerated mobility in the center of the pores, highlighted by a jump in heat capacity in the DSC curve of the confined simvastatin, at lower temperatures than bulk SIM (see Figure 7b).

## 4. Conclusions

Simvastatin, a lowering high-cholesterol drug, was loaded onto MCM-41 mesoporous silica (~3.5 nm average pore diameter) as an approach to stabilize it in the amorphous state and enhance its dissolution in an aqueous medium. To access the physical state of the loaded drug and its in-pore distribution, a variety of experimental techniques was used (PXRD, ssNMR, and ATR-FTIR), showing that SIM is fully amorphous upon loading; however, it is unevenly distributed inside pores. Molecular Dynamics simulations provide evidence for this pore diameter, that most SIM molecules tend to accumulate near the inner pore wall, forming a surface layer anchored at multiple sites through up to five HBs. This is in accordance with both infrared and thermogravimetric analysis, which revealed strong guest–host interactions and a high thermal-resistant population, respectively. Moreover, the multiple binding sites turn the anchored molecules dynamically rigid and, therefore, lacking a calorimetric signature and a resolved response in dielectric relaxation spectroscopy.

MD simulations also predicted a pore core molecular fraction spatially limited to a pore radius ~1 nm that adopts preferential radial orientation of dipoles distinct from bulklike SIM. This is consistent with the detection of a calorimetric glass transition with an onset decreased by around 15 degrees relative to bulk amorphous SIM and considered a manifestation of true confinement effects. Failure to detect a corresponding α_fast_ dielectric process is due to the cancellation of the overall dipole moment. However, intramolecular motions were able to be detected originating: (i) a *γ* -process assigned to dipolar reorientations of the lactone hydroxyl groups, and (ii) a *β* relaxation with better resolution, attributed to rearrangements of the carbonyl carrying ester tail.

The pore-core mobile fraction, first released from the MCM carrier, exhibits a highly enhanced delivery rate compared to neat crystalline SIM dissolution. Furthermore, the narrowness of the pore size prevents drug recrystallization and, therefore, incorporation in MCM proved to be a suitable strategy to improve the release rate and allow the long-term stabilization of SIM in the amorphous state (at least three years after incorporation).

The thorough investigation of SIM:MCM by complementary techniques and methods carried out in this work allowed the fundamental understanding of the target material relevant to the future design of controlled drug delivery systems.

## Data Availability

Not applicable.

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
