# Peer review of "Evidence of Strong Guest–Host Interactions in Simvastatin Loaded in Mesoporous Silica MCM-41"

_pharmaceutics, 2023, doi:10.3390/pharmaceutics15051320_

Round 1

Reviewer 1 Report

The submitted manuscript is very interesting, presents huge amount of new information on the very important API, simvastatin. The Authors have combined a lot of experimental (physiochemical) and theoretical methods in order to get a full insight into the studied composite, bravo! However, it also requires major revision before it can be proceeded further.

Abstract is significantly too long, it should be shortened, i.e. by removing the parts from lines 31-38

In the introduction it should be stated if the MCM-41 can be used as an excipient? I’m very worried about its cytotoxicity.

Line 51, not “science” but “sciences”

Line 64, at this point some information about the polymorphism of SIM would be beneficial

Lines 101-123, this part should be moved to the Discussion as it presents your results. It shouldn’t be the par of Materials and Methods.

Line 156, the recycle delay is quite short. Are you sure the 2s is sufficient to prevent the signal saturation?

Line 210, why not NPT?

Figure 4, what was the refcode of the crystal used for the PXRD simulations? I guess the structure originated from CCDC, yes?

Figure 5, the spectra of amorphous and crystalline SIM should be presented for comparison. If it is impossible for the Authors to record the spectra on their own, which would be best, please copy the spectra from the previous (cited) works. Nowadays, obtaining permission to copy the figure from previous works is quite simple.

ssNMR, those kind of spectra can be obtained when the recycle delay is too short. It would be reasonable to apply longer recycle delay.

Minor editing of English language required

Reviewer 2 Report

The Authors loaded simvastatin, an antihyperlipidemic drug, in a mesoporous silica matrix in order to obtain its stable amorphization which can ensure an improvement of its pharmaceutical behavior. Thus, the issue of this paper is of sure interest for the pharmaceutical researcher since the issues of the low solubility and the instability of the amorphous phase are very common in the pharmaceutical field.

The Authors made an extensive work. They used a number of techniques and made a reasoned discussion of all the results obtained. The text is well written.

There are only few minor revisions that needed to be addressed:

-According to the IUPAC recommendations, for DSC and TGA techniques, the term “curve” should be used instead of “thermogram”.  Thus, please make the necessary changes in the text.

-Table 2 makes no contribution to the text, it can be delated.

Round 2

Reviewer 1 Report

The revised version can be accepted for publication.